# Quality of Life in Patients over Age 65 after Intestinal Ostomy Creation as Treatment of Large Intestine Disease

**DOI:** 10.3390/ijerph20031749

**Published:** 2023-01-18

**Authors:** Joanna Chrobak-Bień, Anna Marciniak, Izabela Kozicka, Anna Lakoma Kuiken, Marcin Włodarczyk, Aleksandra Sobolewska-Włodarczyk, Anna Ignaczak, Ewa Borowiak

**Affiliations:** 1Department of Conservative Nursing, Faculty of Health Sciences, Medical University of Lodz, Jaracza 63, 90-251 Lodz, Poland; 2Nursing, Medical University of Lodz, 90-251 Lodz, Poland; 3Surgical Oncology, Community Hospital, Munster, IN 46321, USA; 4Department of General and Colorectal Surgery, Faculty of Medicine, Medical University of Lodz, 90-251 Lodz, Poland; 5Department of Gastroenterology, Medial University of Lodz, 92-213 Lodz, Poland

**Keywords:** intestinal stoma, quality of life, large intestine

## Abstract

Introduction: For patients with severe intestinal diseases, ostomy surgery can be health-preserving and even lifesaving. Unfortunately, stoma creation also results in a morbidity that patients must manage. Utilization of the correct ostomy appliances is essential for the patient to regain full daily fitness. Patients also now have access to stoma clinics and fistula support groups where they can receive education and emotional support. Aim: The aim of the study was to assess the quality of life of patients over 65 years of age with an intestinal stoma, created for treatment of severe colorectal disease. Material and methods: The study involved 100 patients (52 women, 48 men) over the age of 65 with an intestinal stoma. Demographic and medical information was collected. The patients completed diagnostic surveys using the SF-36v2 questionnaire and the author’s questionnaire. Results: Analysis demonstrated statistically significant relationships between the quality of life of the patient population and stressors of everyday life. Furthermore, there are statistically significant relationships between quality of life and demographic factors including age, marital status, place of residence, and education. Only gender was not a statistically significant factor. Conclusions: A lengthened time interval to intestinal stoma creation is associated with an improved quality of life as well as psychological and emotional acceptance of the intestinal stoma. Support relationships with loved ones is associated with the acceptance of an intestinal stoma. There is a relationship between acceptance of an intestinal stoma and demographic factors such as marital status, place of residence, and education. Gender did not show any significant relationship. Stoma complications are not related to the acceptance of an intestinal stoma.

## 1. Introduction

Quality of life assesses an individual’s perception of all aspects of their everyday life. Assessing quality of life in elderly patients is of particular importance due to the increasing prevalence of comorbidities and increasing dependence on the environment. According to Bosacka et al. [1], the elderly constitutes an isolated group, discriminated against, ignored, and most willingly forgotten. For these reasons, advocates strive to improve and maintain functional physical, cognitive, and emotional fitness in seniors.

In the modern world, the proportion of the older population is increasing day by day due to better health care systems, improved life standards, advances in medicines, and reduction in mortality rate. The physiological changes which occur as a result of aging are very obvious, which led to the development of a special Geriatrics-branch. The term geriatrics deals with the health of elderly people. Common surgical diseases which need surgery in the elderly population are cholelithiasis, abdominal hernias, and various cancers. Many of these conditions present lately in elderly people and are associated with increased complication risks [2].

Chronic disease, cancer, and/or surgery debilitate elderly persons by weakening the body, reducing activity, and reducing adaptability. In particular, colorectal cancer (a highly prevalent malignancy increasing with age) disproportionately affects the elderly population. Treatment strategy depends on the cancer’s stage at diagnosis; surgery is usually required during the treatment course. One surgical option is an intestinal ostomy surgery (i.e., stoma creation surgery), which improves the patient’s prognosis and affects his or her later quality of life [3]. As ostomies are being placed in 35% of surgically treated older patients with colorectal cancer, is important to have an insight into the impact of a stoma on the quality of life in such patients [4,5]. Proper stoma care is also a key element of improving stoma patients’ quality of life. A literature review indicates that patients’ psychological and emotional acceptance of a stoma correlates with positive self-perception, while the lower a patient’s acceptance of a stoma, the greater the patient’s aversion to oneself. Thus, emotional and psychological acceptance is the most important factor in the development and maintenance of emotional disorders in elderly stoma patients [6].

Support for patients with a stoma begins in the hospital ward, where the patient is prepared for self-care at home. Patients receive ostomy care teaching, information and resources on ostomy appliances, and forms for financial reimbursement related to ostomy care. Some patients need assistance from third parties to provide care at home. Family support for patients with stomas is also an important predictor of a patient’s improvement in quality of life and the speed at which the patient adapts to their new situation [7]. 

Patient mental attitude plays an important role in disease treatment. The manner in which a patient assesses their own health condition and potential complications determines their quality of life and influences the way they function in everyday life [8]. Acquiring a stoma dramatically changes a patient’s personal and social life. Furthermore, the patient’s own general state of health and comorbidities have an influence on the adjustment to the intestinal stoma [9]. Adaptation to new living conditions with a stoma is supported by factors such as: (1) acceptance from the partner, (2) return to professional life, (3) support from family and friends, and (4) health protection [6]. The aim of the study was to assess the quality of life of patients over 65 years of age with an intestinal stoma, created for treatment of severe colorectal disease.

## 2. Materials and Methods

The study involved 100 patients (52 women, 48 men) over 65 years of age with an intestinal stoma. The research was carried out from February to May 2022 in the stoma clinic of the Independent Public Healthcare Center of the University Teaching Hospital and Military Medical Academy of the Medical University-Central Veterans Hospital in Lodz, Poland. Participation in the study was voluntary and anonymous, of which the respondents were informed at the beginning of the study. The proposal of this study did not require the approval of the Bioethics Committee, as it does not bear the hallmarks of a medical experiment. The study was conducted in accordance with the guidelines of the Helsinki Declaration. Patients signed informed consent for all the diagnostic and therapeutic procedures during hospitalization. All of the gathered data was confidential.

The study was conducted with the use of the diagnostic survey method, using the standardized SF-36v2 questionnaire (The Short Form (36) Health Survey, version 2), licensed with the QM059021 license, and the author’s questionnaire.

The health-related quality of life was assessed using the Polish-language version of the licensed form SF-36v2 (Student License Agreement QM035225-CT177402-OP052598). The SF36v2 questionnaire consisted of 11 questions which assessed the health-related quality of life in the following individual domains: physical fitness-PF (Physical functioning), activity limitations because of health condition-RP (role limitations due to physical problems), ailments pain-BP (bodily pain), general health perception-GH, vitality-VT (vitality), social functioning-SF (social functioning), mental health-MH (mental health), activity limitations caused by emotional problems-RE (role limitation due to emotional problems), and change of health state-HT (health transition). The above categories were grouped into two component summary scales: physical PCS (Physical component summary), which included: PF, RP, BP, GH, and mental component summary (MCS), which included: VT, SF, RE and MH-mental functioning, total mental health [9,10].

Our own questionnaire consisted of 21 questions concerning disease duration, complications, treatment used thus far, stoma impact on the quality of life, relationships with loved ones after stoma creation, intimate contact after the stoma creation, physical activity after stoma creation, level of acceptance of the current life situation, and demographic data. 

Statistical analysis was performed using the χ^2^ (chi-square) test of independence to compare observed results with expected results. A 5% risk of inference error was assumed. A probability value of *p* < 0.05 was considered statistically significant.

## 3. Results

The group of 100 respondents were people over the age of 65 who had had intestinal ostomy surgery in the past. The detailed characteristics of the study group are presented in Table 1.

For 57% of respondents (*n* = 57), the greatest difficulties in everyday life arise from the care of the skin around the stoma and the replacement of the pouch. Other common inconveniences include leakage of ostomy bags and unpleasant odors reported by 38% of respondents (*n* = 38). Almost every respondent (88%, *n* = 88) empties their ostomy pouch or bag on their own. Only 9% of respondents (*n* = 9) require the help of their relatives in this activity. Half of the respondents (50%, *n* = 50) spend 10 to 20 min taking care of the skin around the stoma together with the ostomy pouch or bag replacement. The small group of 8% (*n* = 8) are people who spend more than 60 min on this activity. The nurse’s role in caring for the intestinal stoma patient is essential and should be emphasized. As shown by the survey, nurses comprise the main source of education, information, and support for ostomy appliances for the respondents (74%, *n* = 74). The least frequent source of help for ostomy care is that from a doctor in 6% (*n* = 6) of respondents and a pharmacist in 3% (*n* = 3). More than half of the respondents, 66% (*n* = 66), confirmed that they had a stoma complication. The most frequently reported complication was the peristomal hernia in 29% of respondents (*n* = 29). The least frequent complications include allergy and dermatitis in 2% of respondents (*n* = 2) and the lack of mucocutaneous union in 2% (*n* = 2). Assuredly 67% (*n* = 67) of respondents list their family as their main source of emotional support, while 17% (*n* = 17) convey that the nursing staff provided the main source of emotional support. The respondents seldom seek support in their own group of their friends. However, the study showed that 35% of respondents (*n* = 35) use support groups for people with an intestinal stoma. Analysis of health-related quality of life scores from SF-36 questionnaire revealed that respondents were most affected by emotional limitations after stoma creation, while pain was the least affecting factor measured (Table 2). 

The general assessment of the physical and mental condition of the study group contained in the SF 36v2 questionnaire is presented in Table 3.

Statistical analysis did not show any significant correlation between the age of the respondents with an established intestinal stoma and their quality of life (Table 4, *p* > 0.05).

However, the results show a significant relationship between respondents’ marital status and their quality of life after intestinal stoma creation. Over 66% of single respondents admit that their quality of life has slightly deteriorated, and over 66% of widows/widowers claim their quality of life has significantly deteriorated. Statistical analysis also shows a significant relationship between the respondents’ place of residence and their quality of life after intestinal stoma creation: 70% (*n* = 26) of respondents living in a city of over 500,000 inhabitants and 62% (*n* = 10) of rural respondents admit that their quality of life has slightly deteriorated.

The study also shows a significant correlation between respondents’ formal educational levels and their quality of life after intestinal stoma creation. Over 67% (*n* = 25) of respondents with vocational education admit that their quality of life has slightly deteriorated. Only people with higher education, 37.5% (*n* = 6), declare that the presence of an intestinal stoma does not affect their quality of life.

Statistical analysis showed a significant relationship between the time that elapsed since intestinal stoma creation and their current quality of life. Sixty percent (60%, *n*= 12) of respondents who had a stoma 2-to-4 years ago admit that their quality of life has slightly deteriorated. In this same group of respondents with a stoma placed 2-to-4 years ago, 25% (*n* = 5), believe that the presence of a stoma does not affect their quality of life. On the other hand, 59% (*n* = 23) of respondents with a stoma for less than 6 months say that their quality of life has significantly worsened. Thus, the longer the interval between the stoma creation and point of survey, the less impact of the stoma on the quality of life.

More than 59% (*n* = 16) of the respondents who underwent stoma creation 7-to-12 months ago partially accept the intestinal stoma emotionally and psychologically, while more than 30% (*n* = 12) of the respondents who have had a stoma for less than 6 months do not accept the intestinal stoma at all. Thus, again, the longer the interval between stoma creation and point of survey, the more acceptance of the intestinal stoma itself.

Statistical analysis showed a significant relationship between declining quality of relationships with loved ones and the patient’s acceptance of an intestinal stoma. Over 54% (*n* = 6) of the respondents experiencing deteriorating relationships with their loved ones accept the stoma only partially. However, over 27% (*n* = 12) of the respondents who have not changed their relationships with their loved ones fully accept the intestinal stoma.

The study showed a significant relationship between the age of the respondents and what causes the greatest difficulties for them in everyday life. All of the respondents aged 77–81 (100%, *n* = 11) admit that they find it difficult to care for the skin around the intestinal stoma and replace the pouch or bag. Over 63% of respondents aged 71–76 (*n* = 17) admit that they find it difficult to open the ostomy bags and their bags emit unpleasant odors. A significant relationship was also shown between the place of residence of the respondents and what causes the greatest difficulties for them in everyday life. Over 81% (*n* = 13) of respondents living in the countryside admit that they find it difficult to care for the skin around the intestinal stoma and replace the pouch, while over 57% (*n* = 24) of respondents living in a city with up to 500,000 inhabitants say that the difficulty is in opening the ostomy bags, which give off an unpleasant odor.

The study showed a significant relationship between the place of residence of the respondents and the acceptance of an intestinal stoma in the current situation. Over 68% of respondents (*n* = 11) living in rural areas partially accept the intestinal stoma, but over 25% of respondents (*n* = 12) living in cities with up to 500,000 inhabitants do not accept an intestinal stoma (Table 5).

Statistical analysis also showed a significant relationship between the respondents’ formal education level and what in everyday life causes them the greatest difficulties. Most respondents with vocational education (80%, *n* = 28) admit that they find it difficult to care for the skin around the intestinal stoma and replace the pouch. Around 50% of respondents with (*n* = 15) secondary education claim that they find it difficult to open the ostomy bags, which emit an unpleasant odor. However, the study did not show a significant relationship between the sex of the respondents (Table 6), age, and acceptance of an intestinal stoma in the current situation.

Statistical analysis of the results obtained in the study did not show a significant relationship between the occurrence of stoma complications and the acceptance of an intestinal stoma in the current situation (Table 7).

## 4. Discussion

Colorectal cancer is a common affliction, most often in people over 50 years of age [11]. For the elderly, the stoma can become problematic if they are not sufficiently independent, live alone, and have limited, if any, outside support. According to the literature review, it is most often women and people over 60 who struggle to psychologically and emotionally accept intestinal stomas [12].

Studies demonstrate that 90% of respondents experience a change in their quality of life after creation of their intestinal stoma. Demographic factors such as age, marital status, place of residence, and education also influence quality of life. Research indicates that quality of life deteriorates significantly with age, despite the differing views put forth by Glińska et al. [13].

Data suggests that people aged 64-to-70 adapt better to life with a stoma [14]. A total of 60% of people who had an intestinal stoma 2-to-4 years ago say their quality of life slightly worsened. According to Dziedzic et al. [14], there is a relationship between the time that has elapsed since the creation of the stoma and their current attitude. The authors proved that the more time passed after the surgery, the calmer the respondents, and the better they coped with stoma creation. Lewandowska et al. [15] believe that the more time that has passed since the emergence of an intestinal stoma, the more resourceful the patients become. Starczewska et al. [7] confirm the relationship between time and quality of life in nearly all domains of life (i.e., physical functioning, physical limitations, pain ailments, mental health, vitality, social functions, and general health), except the domain of the emotional state. The longer the patient has a stoma, the higher his quality of life [7]. In the studies by Szymańska-Pomorska et al. [16], the fear of contamination (32.1%), lack of control over gas and stool (29.2%), or bad smell (27.3%) caused the most difficulties. However, our own research showed that the patients’ main problem in everyday life is caring for the skin around the stoma together with ostomy bag replacement (57%), followed by leakage of ostomy bags, and unpleasant odors (38%). This may indicate that manual dexterity decreases with age and stoma activities may be more difficult for the elderly than for younger people.

According to the obtained results, as many as 42% of respondents seek help from the closest person to change their ostomy appliances. It goes without saying that stoma complications also make it difficult to manage the intestinal stoma; in such instances, the help of relatives or medical staff is inevitable. Piaszczyk and Schabowski [17] believe that the care and proper replacement of the ostomy pouch causes significant problems for patients which often require external assistance. In addition, Rogowska et al. [18] believe that the time devoted to hygiene and care activities of the stoma usually ranges from 20-to-30 min. Based on the data provided, one can reasonably conclude that the elderly are most likely to try to apply the ostomy pouch carefully or repeatedly attempt to apply the ostomy pouch. 

Periostomy complications intrinsically worsen patients’ adaptation to everyday life [12]. Periostomy complications make it difficult to care for a stoma and apply an ostomy pouch. In conducted studies, 66% of respondents confirmed the occurrence of stoma complications. A common complication, occurring in 29% of respondents, was a peristomal hernia. The respondents also reported the occurrence of skin chafing, peristomal fistulas, collapsed stoma, and ulcerations. The analyses presented by Bielecki [19] show that a peristomal hernia appears over time in almost every patient with an intestinal stoma. Early stoma complications stem from incorrect stoma site positioning or poor stoma formation during surgery.

In our own study, we found patients usually get teaching and informational support (educational resources) regarding ostomy appliances from a nurse (74%), and the source of emotional support is most often the family (67%). Woźniak and Egzuszna-Owcarz [20] obtained similar results after conducting research in a proctology clinic in Płock, where over 60% of patients receive informational support from a nurse, while 73.8% receive emotional support from their family. Nurses are an integral part of the treatment process, as is the patient’s family. Positive relationships with loved ones influence the acceptance of the intestinal stoma in full.

Our research found that 65% of respondents do not use support groups for people with an intestinal stoma; of these, 33% did not know about the existence of the support groups. Instead, most patients (82% of the respondents) visited the stoma clinic more often. Research by Lewandowska et al. [15] showed that 96% of respondents do not use support groups, while 88% of people utilize help from an ostomy clinic. Gęsicka et al. [21] also confirms a very small number of people identified as belonging to a support group or association. Notably, people using support groups or associations assess their quality of life at a good level [18]. Patient education is extremely important at the stage of adapting to everyday life. It covers many aspects related to stomas including stoma care, the fitting of appropriate ostomy equipment, emotional and informational support, education of the patient’s families, and preparation for conscious self-care. In addition, aging itself brings increasing limitations in various aspects that in themselves comprise quality of life and daily functioning, irrespective of the stoma [22,23]. Moreover, this group of patients might be less demanding, a phenomenon which is referred to as ‘response shift’ [24,25]. This is an internal psychological process of change in standards, values, or conceptualization of quality of life over time. As a result, required changes in lifestyle and problems due to the ostomy might be limited or experienced in a different way. Specific preoperative stoma education aimed at the issues faced by elderly stoma patients might facilitate acceptance of the ostomy and could limit the occurrence of stoma-related problems and functional limitations even further [26].

The analysis of the SF-36v2 questionnaire revealed that the average overall assessment of physical health is 46.47 ± 15 and the average overall assessment of mental health is 49.65 ± 19. Studies by Bayar et al. [27] show that the average general quality of life in patients aged 20-to-54 was 41, suggesting a poor quality of life. This may be due to the relatively young group of respondents. According to the authors of the study, young people more often do not accept their intestinal stoma [27].

This study has some limitations. First of all, there is a risk of selection bias of both patients and normative respondents. Patients who responded in our study to the questionnaire were fit enough and willing to participate in research. The normative group is a slightly different group with minor baseline differences, especially in educational level, although we corrected for this potential confounder in the analyses. Finally, it is likely that the type of stoma affects the quality of life, but the type of stoma was registered in only 15% of the patients and was not included in the results. Future research could focus on these data and correct for this (potential) confounder. Despite these limitations, this is one of the first studies to focus on the comprehensive impact of stomas on elderly patients treated for large bowel diseases.

In summary, we can conclude that quality of life of people over 65 with an intestinal stoma is average, but for many, results in a low quality of life. Notably, the later in life an intestinal stoma is created, and the shorter the time interval from surgery, the quality of life is lower. Unfortunately, previous research, which is limited, demonstrates this phenomenon. It is impossible to ignore the impact of accompanying diseases, mental disorders, or physical limitations on the daily functioning of people with intestinal stomas. For example, 85% of respondents report that their physical activity has deteriorated. However, the family most often takes part in caring for the patient, as the patient himself does not necessarily cope with stoma challenges, e.g., pouch replacement. Support from family and loved ones improves quality of life. It has a positive effect on acceptance of the disease and proper care of the intestinal stoma, just as utilization of stoma clinics. The patient’s family or relatives involved in caring for the patient facilitate the patient’s return to everyday life. Thanks to such support and assistance, it is easier to prevent adverse events, and detect health-related irregularities earlier so that medical attention can be given as soon as possible.

## 5. Conclusions

A lengthened time interval to intestinal stoma creation is associated with improved quality of life and acceptance of an intestinal stoma.Supportive relationships with loved ones is associated with the acceptance of an intestinal stoma.There is a relationship between quality of life and demographic factors, such as: age, marital status, place of residence, education, as well as the greatest stressors in everyday life. Gender is not a statistically significant factor.According to the SF-36v2 questionnaire, the impact of age on the level of quality of life in patients with an intestinal stoma is still uncertain. Future research could focus on these data and focus on how the risk of complications impacts the quality of life in this group of patients.There is a relationship between acceptance of an intestinal stoma and demographic factors such as: marital status, place of residence, and education. Gender and age did not show any significant association with quality of life.Stoma complications are not related to the acceptance of an intestinal stoma.

## Figures and Tables

**Table 1 ijerph-20-01749-t001:** Characteristics of the studied group.

Characteristic		*n*	%
Gender	woman	52	52
	man	48	48
Age	65–70 years	44	44
	71–76 years	33	33
	77–81 years	13	13
	82–89 years	5	5
	90 > years	5	5
Marital status	never married / bachelor	9	9
	divorced / divorced	6	6
	widow / widower	24	24
	married / married / in a partnership	61	61
Domicile	city of >500 thousand inhabited	37	37
	city up to 500 thousand inhabited	47	47
	village	16	16
Education	basic	7	7
	medium	40	40
	higher	16	16
	professional	37	37
Time since stoma creation	<6 months ago	39	39
	7–12 months ago	27	24
	2–4 years ago	20	20
	>5 years ago	14	14
Stoma creation and the impact on the quality of life	did not affect the quality of life	10	10
	yes, it slightly worsened the quality of life	43	43
	yes, it has greatly worsened the quality of life	47	47
Relationships with loved ones after stoma creation	have not changed	43	43
	more support than before the procedure	46	46
	less support than before the procedure	11	11
Intimate contact after the stoma creation	intimacy has not changed	71	71
	yes, I avoid intimate contacts	29	29
Physical activity after stoma creation	has not changed	12	12
	yes, for the worse	85	85
	yes, for the better	3	3
Acceptance of the current life situation	I accept the intestinal stoma partially	43	43
	I accept the intestinal stoma fully	24	24
	I do not accept the intestinal stoma	18	18
	other	15	15

*n*—total number of patients.

**Table 2 ijerph-20-01749-t002:** The descriptive statistics for SF-36 health dimensions.

Descriptive Statistics	*n*	Mean	SD
Physical functioning	100	30.4	17.3
Physical limitations	100	81.8	59.1
Emotional role functioning	100	90.7	61.5
Bodily pain	100	29.0	33.5
Mental health	100	35.8	9.4
Vitality	100	35.5	11.0
Social role functioning	100	36.6	10.0
General health	100	44.7	10.8

*n*—number of observations; SD—standard deviation.

**Table 3 ijerph-20-01749-t003:** The summary measures from the SF-36v2 questionnaire.

Descriptive Statistics	*n*	Mean	SD
Overall assessment of physical health	100	46.47	14.52
Overall mental health assessment	100	49.65	19.45

*n*—number of observations; SD—standard deviation.

**Table 4 ijerph-20-01749-t004:** Quality of life using the SF-36v2 questionnaire and age.

Age Range		Overall Assessment of Physical Health	Overall Mental Health Assessment
≥90 years	Mean	50.0583	46.5833
	N	5	5
	SD	20.21337	18.02631
65–70 years	Mean	48.8930	54.5312
	N	44	44
	SD	14.69540	19.65403
71–76 years	Mean	41.5619	42.0770
	N	33	33
	SD	12.83511	17.33432
77–81 years	Mean	49.1699	53.0609
	N	13	13
	SD	15.82839	19.62123
82–89 years	Mean	46.8500	50.8750
	N	5	5
	SD	10.62102	23.34003
Overall	Mean	46.4658	49.6500
	N	100	100
	SD	14.51676	19.45490
F		1.466	2.177
*p*		0.219	0.077

*n*—number of observations; *p*—level of statistical significance; SD—standard deviation; F—result of the ANOVA test (analysis of variance).

**Table 5 ijerph-20-01749-t005:** Intestinal stoma acceptance and place of residence.

			What Is Your Mental Attitude in the Current Situation?				Overall	Pearson Chi-Square	*p*
			I Accept an Intestinal Stoma Partially	I Accept an Intestinal Stoma Fully	Other	I Do Not Accept an Intestinal Stoma			
Please select your place of residence:	a city with 500 thousand inhabitants and more	*n*	11	12	8	6	37	14.99	0.02
		%	29.7%	32.4%	21.6%	16.2%	100,0%		
	city with up to 500 thousand inhabitants	*n*	21	7	7	12	47		
		%	44.7%	14.9%	14.9%	25.5%	100.0%		
	village	*n*	11	5	0	0	16		
		%	68.8%	31.3%	0.0%	0.0%	100.0%		
Overall		*n*	43	24	15	18	100		
		%	43.0%	24.0%	15.0%	18.0%	100.0%		

*n*—number of observations; *p*—level of statistical significance; χ^2^—Persona’s chi square test result.

**Table 6 ijerph-20-01749-t006:** Intestinal stoma acceptance and gender.

			What Is Your Mental Attitude in the Current Situation?				Overall	Pearson Chi-Square	*p*
			I Accept an Intestinal Stoma Partially	I Accept an Intestinal Stoma Fully	Other	I Do Not Accept an Intestinal Stoma			
Please select your gender:	woman	*n*	19	16	5	12	52	6.76	0.8
		%	36.5%	30.8%	9.6%	23.1%	100.0%		
	man	*n*	24	8	10	6	48		
		%	50.0%	16.7%	20.8%	12.5%	100.0%		
Overall		*n*	43	24	15	18	100		
		%	43.0%	24.0%	15.0%	18.0%	100.0%		

*n*—number of observations; *p*—level of statistical significance; χ^2^—Persona’s chi square test result.

**Table 7 ijerph-20-01749-t007:** Intestinal stoma acceptance and the incidence of stoma complications.

			What Is Your Mental Attitude in the Current Situation?				Overall	Pearson Chi-Square	*p*
			I Accept an Intestinal Stoma Partially	I Accept an Intestinal Stoma Fully	Other	I Do Not Accept an Intestinal Stoma			
Have you experienced any stoma complications?	no	*n*	17	7	5	5	34	1.15	0.765
		%	50.0%	20.6%	14.7%	14.7%	100.0%		
	yes	*n*	26	17	10	13	66		
		%	39.4%	25.8%	15.2%	19.7%	100.0%		
Overall		*n*	43	24	15	18	100		
		%	43.0%	24.0%	15.0%	18.0%	100.0%		

*n*—number of observations; *p*—level of statistical significance; χ^2^—Persona’s chi square test result.

## Data Availability

Not applicable.

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
