# Peer review of "Quality of Life in Patients over Age 65 after Intestinal Ostomy Creation as Treatment of Large Intestine Disease"

_ijerph, 2023, doi:10.3390/ijerph20031749_

Round 1

Reviewer 1 Report

First of all, I would like to congratulate you on your work. I think it is a very relevant and much needed study, as unfortunately more and more people have to live with a stoma every day.

Introduction

Overall it is a good introduction, which completely focuses the reader on the topic under study. However, I think that some more information could be included between the first paragraph and the second paragraph on the rise of chronic diseases, as it shows a disconnect between the first idea and the second. This would make for a more coherent exposition of the issue. It would also be interesting to provide data on how many people who are diagnosed with cancer end up with a stoma. I also think that more emphasis should be placed on the fact that information and support for ostomates begins before surgery, as this is vital for subsequent adaptation. Finally, a reference is missing at the end of the article, as they state 4 points on which acceptance is based, but do not cite the manual or study on which they base this statement.

Material and Methods

In general, it is well presented but there are small details missing that should be included. For example, they talk about 100 patients with ostomies, but it is not clear to me whether it is a colostomy or an ileostomy. This is important because, as you know, the management of a colostomy is easier than the management of an ileostomy. On the other hand, there is a lack of data on the sample: how many people were invited to participate, and did they all agree to participate? If not, why not? Furthermore, there is no information on who, how and where the patients were recruited and who, how and where the interviews were carried out. Were they self-administered? Were they carried out on-line? If so, was there prior training where the project was explained and any possible doubts that might arise on the part of the participants were resolved?

On the other hand, with respect to the sample, what inclusion and exclusion criteria were used? Was there any kind of randomisation or was it a convenience sample? This is important to know as it will greatly influence the interpretation of the results.

As for the questionnaires, I think that both the choice of questionnaires and their explanation are very good, but I have doubts about the self-developed questionnaire that you present in your study. Could you add something more about it, how was it carried out? What was the basis for its elaboration? Who was or were involved in its elaboration?

Results

Although the results are clearly presented, I think that you should make some changes to them, as there are items that cannot be compared with each other, because they are not real. The difference in the number of people with higher education or even people living in rural areas is an important bias, which could perhaps be resolved with appropriate inclusion and exclusion criteria. I would also like to include an item on the body image disorder caused by living with a stoma. I believe that this is an important issue to address since, despite the fact that we are dealing with people over 65 years of age, image is always important for the work on self-concept and the improvement of quality of life.

Discussion

The discussion is well worked out, but I am concerned about certain aspects. It is clear that, according to their results, the longer a person has a stoma, the more acceptance they will have, because they obviously have to live like this. But perhaps they should go deeper into this idea, go more deeply into this acceptance of the new situation and not ignore it, as there are many theories that explain this result. I also believe that their work would be more robust if they compared other studies carried out on other pathologies with reference to quality of life. In the end, many acceptance processes are similar in different pathologies with very similar consequences to living with a stoma.

It would also be interesting to expand on the issue of support, as your results do not say where this support comes from, from health professionals, from family members, etc. I think it would add more depth to the study. I think it would add more depth to the discussion of the results they present.

In general, I think that the discussion should be worked on more. It seems that they have only limited themselves to presenting their results without going into them in depth, and they are very interesting results that would require a much more in-depth discussion.

Finally, I think that a section on limitations is missing, including, for example, the type of stoma we are talking about, the limitation of sample recruitment... among others.

Conclusions

Some of the points are not clear from their results, they should rewrite them, taking into account their work. In addition, point 4 is very ambiguous and is in contraction with a study cited in your own discussion.

Author Response

The authors would like to thank the Reviewer#1 for their valuable comments and suggestions. Manuscript has been revised and all necessary changes has been done in text. The linguistic mistakes have been corrected. The introduction and discussion sections have been improved accordingly to comments. The recent studies form the period 2018-2022 were added and discussed to improve the paper.

Reviewer 2 Report

Review of:  ijerph-1970771

Regard the topic:, unfortunately, after careful consideration my opinion is that this paper is not suitable for publication in the Special Issue: Health Promotion and Quality of Life among Older Adults because this is clinical not public health research. Considering specific topic, probably some other journal with the aim&scope more into health related QOL of cancer patients will be more appropriate for publishing such a paper. This Journal has accent on public health, population issues.

There are some methodological issues that need to be solved before further considering this paper.

Suggested improvements and additional comments on the tables and figures:

Ln. 74 and 76 The study…  repeated sentence. Correct the paragraph – delete repeated sentence.

Please note that SF-36 questionnaire measures health related QOL, not QOL in the narrower sense, and it would be good to mention in the description that Health related QOL was measured.

Table 1 – " Stoma creation and the impact on the quality of life", "Relationships with loved ones after stoma creation"

are those separate, additional questions in the study or was it derived from the SF36 results? If it is additional items/questions in survey, this needs to be described in methods section.

Table 2 – the names of the SF-36 dimensions do not correspond to original English dimension names (no "sanity" dimension, see SF-36 manual, this dimension supposed to be named: mental health dimension, "pain ailments" also, do not correspond to original name). For example, in Ln.211 dimension mental health is correctly named.

-          Table title need to be changed in order to describe what table present (descriptive statistics for SF-36 health dimensions)

Table 3 – table presents SF-36 summary measures, title need to be changed to describe that.  Dimensions names write in accordance to the names in Manual.

Quality of life indicator as a single score can not be calculated form SF-36 questionnaire (see the manual for original questionnaire). It is not acceptable for authors to introduce own calculation of the scores from the standardised, validated, authorised questionnaire.

Table 4 – how is calculated Quality of life indicator? This need to be described in methods since it is presented in results and analyzed as a variable.

Ln. 90 - The above categories are grouped into two main scales…  according to the original SF-36 those are:  The physical (PCS) and mental (MCS) component summary scales, not the main scales

Ln. 122 - Quality of life analysis found that….  It is advisable to say: Analysis of the SF-36 results or Analysis of health related QOL scores from SF-36 questionnaire revealed….

Ln. 134  - instead of “study shows” it is advisable to say “the results show…”

What is the measure of QOL? Single score calculated - how?  SF-36 questionnaire does not allow calculation of single score. The most correct way of presenting SF-36 results is as scores on eight dimensions and after that as a two(2) summary measures which are physical and mental summary scale measure. SF-36   is primarily health status questionnaire and can be used to assess health related functioning, not quality of life in general.

All analysis which include single score derived from SF-36 need to be omitted, and analysis performed in accordance with the SF-36 score analysis guidance. Consequently, some conclusions will be changed.

Ln.159  - How quality of relationships with loved ones was measured?  SF-36 does not include such variable. If there are additional questions, please specify and describe in methodology section. In Ln. 95, after mentioning own questionnaire describe and connect with what is presented in table 1.

Ln.256 – physical activity deteriorated… is this according to SF-36 results or this is separate question? If it is additional question on physical activity, this needs to be stated in Methods section.

 Literature/ references:

It is strongly advised to include English language articles from international journals. In Methods section add the reference for the SF-36v2 manual, and the reference for Polish version of SF-36 which was used (if there is published paper on Polish version construction and validation).

What specific improvements could the authors consider regarding the methodology?

Consult official web site for SF-36 questionnaire regard the terminology, dimension names in English language and computing the score. Take in the consideration difference between SF-36 and SF-36 ver.2 in calculating and presenting dimension scores

https://www.qualitymetric.com/health-surveys/the-sf-36v2-health-survey/

https://www.qualitymetric.com/about/news/scoring-the-sf-36v2-sf-12v2-health-surveys-ranges-for-the-standard-acute-versions/

Regard the SF-36 interpretation it is advisable to consult the following articles:

-          Hays RD, Morales LS. The RAND-36 measure of health-related quality of life. Ann Med. 2001 Jul;33(5):350-7. doi: 10.3109/07853890109002089.

-           Hawthorne G, Osborne RH, Taylor A, Sansoni J. The SF36 Version 2: critical analyses of population weights, scoring algorithms and population norms. Qual Life Res. 2007 May;16(4):661-73. doi: 10.1007/s11136-006-9154-4. Epub 2007 Feb 1.

Are the references appropriate?

It is mandatory to supplement reference list with more English language articles from international journals (Journals in WoS journal list).

it will be beneficial to refer to articles on QOL of elderly and cancer patients, for example (but only this):

-          Di Maio, M., Perrone, F. Quality of Life in elderly patients with cancer. Health Qual Life Outcomes 1, 44 (2003). https://doi.org/10.1186/1477-7525-1-44

-          Wedding U, Pientka L, Höffken K. Quality-of-life in elderly patients with cancer: a short review. Eur J Cancer. 2007 Oct;43(15):2203-10. doi: 10.1016/j.ejca.2007.06.001. Epub 2007 Jul 26.

Author Response

The authors would like to thank the Reviewer#2 for their valuable remarks and suggestions. The manuscript has been revised carefully point-by-point and all necessary correction have been made. We revised the paper according to recent studies from the and improved the paper accordingly the suggestions, the list of references has been updated. The analysis of SF36 Version 2 have been corrected accordingly the comments and improved, the conclusions have been modified accordingly. The limitations of paper have been added.